# Experiences of children's self-wetting (including urinary incontinence) in Cox's Bazar's Rohingya refugee camps, Bangladesh

Mahbub-Ul Alam[1,2⊙], Sudipta Das Gupta[1], Claire Rosato-Scott[2], Dewan Muhammad Shoaib[1], Asmaul Husna Ritu[1], Rifat Nowshin[1], Md Assaduzzaman Rahat[1], Nowshad Akram[3], Joanne Rose[4], Barbara E. Evans[2], Dani J. Barrington[2,5⊙] *

1 Environmental Health and WASH Unit, Health Systems and Population Studies Division, International Center for Diarrhoeal Disease Research, Bangladesh (icddr,b), Dhaka, Bangladesh, 2 School of Civil Engineering, University of Leeds, Leeds, United Kingdom, 3 World Vision Bangladesh, Cox's Bazar, Bangladesh, 4 Department of Health Science, University of York, York, United Kingdom, 5 School of Population and Global Health, The University of Western Australia, Perth, Australia

⊙ These authors contributed equally to this work.
* dani.barrington@uwa.edu.au

**Data Availability Statement:** There are ethical restrictions imposed by the Research Ethics Committee, Faculty of Engineering, University of

## Abstract

Self-wetting is the leakage of urine, either due to the medical condition of urinary incontinence (UI), or because a person does not want to, or cannot, access a toileting facility in time. This study explored the attitudes towards self-wetting and experiences of children (aged five to 11), their caregivers, community leaders and humanitarian practitioners in the Rohingya refugee camps in Cox's Bazar, Bangladesh. We particularly focused on how water, sanitation and hygiene (WASH) and protection interventions might assist in improving these experiences. We purposively selected participants from two camps where our partner organisation works. We conducted Key Informant Interviews (KIIs) with community leaders and camp officials, Story Book (SB) sessions with Rohingya children and in-depth Interviews (IDIs) with caregivers of children who participated in the SB sessions, as well as surveying communal toilets. Self-wetting by children was common and resulted in them feeling embarrassed, upset and uncomfortable, and frightened to use the toilet at night; many children also indicated that they would be punished by their caregivers for self-wetting. Key informants indicated that caregivers have difficulty handling children's self-wetting due to a limited amount of clothing, pillows, and blankets, and difficulty cleaning these items. It was evident that the available toilets are often not appropriate and/or accessible for children. Children in the Rohingya camps appear to self-wet due to both the medical condition of UI and because the sanitation facilities are inappropriate. They are teased by their peers and punished by their caregivers. Although WASH and protection practitioners are unable to drastically alter camp conditions or treat UI, the lives of children who self-wet in these camps could likely be improved by increasing awareness on self-wetting to decrease stigma and ease the concerns of caregivers, increasing the number of child-friendly toilets and increasing the provision of continence management materials.

Leeds, United Kingdom which prevent the public sharing of sensitive minimal data for this study. Data are available upon request from the Research Ethics Committee, Faculty of Engineering, University of Leeds, United Kingdom via email (MEECResearchEthics@leeds.ac.uk) for researchers who meet the criteria for access to confidential data.

**Funding:** The project 'Understanding children and their caregivers' experiences with incontinence in humanitarian contexts' (Project #45432, Principal Investigator DJB) was funded and supported by Elrha's Humanitarian Innovation Fund (HIF) programme, a grant-making facility which improves outcomes for people affected by humanitarian crises by identifying, nurturing, and sharing more effective, innovative, and scalable solutions. Elrha's HIF is funded by aid from the Netherlands Ministry of Foreign Affairs. The funders had no role in study design, data collection and analysis, decision to publish, or preparation of the manuscript.

**Competing interests:** I have read the journal's policy and the authors of this manuscript have the following competing interests: Mahbub-Ul Alam is currently serving on the editorial board of PLOS Global Public Health. This does not alter our adherence to PLOS policies on sharing data and materials.

## Introduction

The medical condition of urinary incontinence (UI) is defined as the involuntary loss of urine, and it is a symptom of a range of health issues (UI can have physical, mental and emotional causes) [1]. It is a complex global health issue that has a negative impact on people's security, dignity, rights and general quality of life [2,3]. In addition to UI, sometimes people 'wet' themselves because they do not want to, or cannot, access a toileting facility in time; not due to a physiological or psychological inability to stop urine leakage. This is sometimes known as social urinary incontinence (SUI). People who leak urine, and their caregivers, face challenges in their everyday lives, with individuals reporting that the intensity and complexity of the experience changes on a daily basis. Incontinence can also result in social and economic marginalization, debilitation, and psychosocial problems due to the associated stigma. The stigmatization of urine leakage prohibits individuals from sharing their difficulties with others, and because of that, they often separate themselves from society, community, and family [4].

Children aged between five and 11 years occasionally wet themselves as a result of the medical condition of UI, and can also experience SUI. Throughout this manuscript, we use the term 'self-wetting' when a child experiences urine leakage, regardless of whether this is UI or SUI (as both of these appeared to be occurring in the study population, and although we asked participants what they believe causes urine leakage, our methods did not allow for medical diagnoses). Regardless of whether self-wetting is physiological, psychological, or due to a lack of appropriate facilities, the shame and humiliation linked with it can have an impact on relationships and involvement in social events, increasing the likelihood of psychological difficulties in childhood. Increased domestic violence towards children who experience UI during sleep has been observed [5] and children who self-wet can also suffer from skin rashes and Urinary Tract Infections (UTIs) [6]. The consequences of self-wetting mean that children who self-wet usually try to conceal the condition, although daytime UI is difficult to conceal, particularly where continence aids such as absorbent pads are not available [7]. Children who do not achieve continence 'on time' (according to culturally specific expectations of achieving toilet training) can suffer long-term psychological issues [7].

Very little work has investigated the prevalence and experiences of self-wetting in low resource settings [3,8], but the available data suggest that it is common around the world, regardless of culture, age and ethnicity [9–12], and that experiences in these settings are poor [13–16]. To the best of our knowledge, the prevalence of childhood self-wetting, including UI, has not been successfully measured in low resource settings, with a recent study that attempted to do so noting that the stigma associated with self-wetting makes it a challenging task to identify children who experience it [17].

In humanitarian settings, self-wetting by all age groups often goes unnoticed [16]. Yet the incidence of UI may be higher in humanitarian settings because of situation-induced trauma, anxiety and physical harm [18,19], and self-wetting generally due to the decreased lack of access to appropriate toileting facilities. For example, International Rescue Committee (IRC), Médecins Sans Frontières (MSF), International Federation of Red Cross and Red Crescent Societies (IFRC) and the United Nations High Commissioner for Refugees (UNHCR) have observed an increase in bed-wetting among children in Syria, Lebanon, Iraq, Greece and Honduras since the onset of fighting and/or their displacement [16]. Jurkovic, et al. [19] found that refugee or displaced children have an increased risk of incontinence, which can cause further trauma in itself. Families in emergency situations face additional challenges in managing their children's self-wetting due to a lack of resources, including water and soap [16]. Self-wetting limits the accessibility of essential services (food, water, and health care) and the opportunity

to participate effectively in decision-making processes, leading to further social marginalization and vulnerability [3,8,16].

There is a lack of knowledge on the challenges and obstacles that children who self-wet and their caregivers encounter in emergency settings, and how support to manage the condition can be effectively provided in the planning, implementation and assessment of humanitarian programming. Self-wetting is a sensitive topic; the research team had to consider carefully whether it was appropriate to include children as participants in such a study. The process of consideration and development of a methodology that was appropriate for vulnerable children are detailed in S1 Text. Ultimately, we decided to include children as they have a right to be heard in research which concerns their personal experiences, and because across the research team we had the necessary expertise to conduct the work in a sensitive and protective manner [20–25].

Our study took a phenomenological approach using a Story Book (SB) methodology [26], in-depth interviews and a sanitation survey to understand the barriers to inclusion and well-being that those living with self-wetting, particularly children aged five to 11 and their caregivers, face in the Rohingya refugee camps of Cox's Bazar, and how more holistic, effective and inclusive WASH and protection programming can be developed to support those who self-wet and their families.

## Methods

We report this study in line with the consolidated criteria for reporting qualitative research [27] (checklist included as S1 Checklist).

### Study setting

This study was conducted in the Rohingya refugee camps of Cox's Bazar, Bangladesh, the world's largest refugee settlement, inhabited by over 950,000 refugees/Forcibly Displaced Myanmar Nationals. Children comprise more than half of the population, with those aged five to 11 years comprising 11% [28]. A concurrent study using the same methodology was conducted in refugee settlements in the Adjumani District of Uganda; results of that study are forthcoming.

### Study design

We used a phenomenological approach to understand lived experiences of children's self-wetting in this context. Although our research design focused on self-wetting with regards urine, some children and caregivers did choose to share their experiences with faecal self-wetting. We undertook SB sessions with Rohingya children aged five to eleven years old, in-depth interviews (IDIs) with their caregivers, key informant interviews (KIIs) with community members (camp leaders, religious leaders and traditional healers) and camp officials (teachers, community health workers, child protection officers and WASH specialists) and surveyed sanitation facilities used by the children and their family members. We then triangulated across these methods to understand lived experiences and possible ways to improve them.

The SB methodology was developed by the research team to hear from children aged five to 11 years old about how an imaginary 'hero' character, approximately their age and living in one of the Cox's Bazar camps, might experience self-wetting. Children were asked to express their understanding, experiences and feelings of facing self-wetting issues through their drawings and discussions. Development of the methodology is outlined in S1 Text. The SB approach was developed using a participatory process with local contextualization and is further discussed and evaluated elsewhere [26]. A process was in place to refer children to child

protection services in the camps if any indicated psychological distress. The IDIs with caregivers and KIIs focused on understanding the experiences of children who self-wet and their caregivers, as well as providing opportunities for participants to provide suggestions on how self-wetting might be prevented. The sanitary surveys evaluated the toileting facilities used by the children in the study for their appropriateness to children of this age group using the CHILD-SAN approach [29].

All of the data collection tools were developed by the research team and Advisory Committee members (specialists on research with children, incontinence and emergency contexts) and then reviewed and finalized in coordination with the local research team in Bangladesh. Questions were translated into Bangla and then verbally adapted into the Rohingya language while conducting activities. All of the tools used (i.e., guidance on conducting the SB approach, interview guides, sanitary surveys) are available in English and Bangla via the Open Science Framework [30] (www.doi.org/10.17605/OSF.IO/E3KHV).

## Participant recruitment

Two research sites (Site 1 and Site 2) were selected based on accessibility, availability of the children and caregivers and pre-existing relationships between the local research partner World Vision Bangladesh—Cox's Bazar (WVB-CXB) and the Rohingya community. Children known to self-wet were not purposively recruited, as the local research team believed this may increase stigma and potentially protection risks. Instead, interested children already known to WVB-CXB (and who may or may not self-wet) were invited to participate during a community visit, with the understanding that even children who do not self-wet may have insights into how those who do are treated (including by themselves) within their community. We conducted eight SB sessions with 48 children, one at each research site with girls aged five to seven years old, boys aged five to seven years old, girls aged eight to 11 years old, and boys aged eight to 11 years old. To understand the caregivers' experiences and understanding of self-wetting by children, we invited (face to face) 12 caregivers of children the SB moderator identified as likely experiencing self-wetting and a further 12 who likely were not, and interviewed them. These 24 caregivers were purposively selected from the eight SB sessions to represent children of different age groups and gender. To further understand the challenges of self-wetting in children and discuss possible solutions to issues raised during the IDIs and SB sessions, we conducted KIIs with individuals who are engaged in the day-to-day care of children living with self-wetting and/or directly involved in addressing issues of self-wetting based on their positions or roles. We invited three teachers, two Community Health Workers (CHWs), three camp leaders, one religious leader, two traditional healers, two child protection officers, and two WASH specialists face to face, by emails or by phone calls, and interviewed them as key informants (Table 1).

## Research training

DJB (PhD, female, Lecturer in Global Health) and CRS (MSc, female, PhD candidate) led the project across both Uganda and Bangladesh, and recruited MUA (MPH and MSS, male, associate scientist) to oversee research training and data collection in Bangladesh due to his experience as a qualitative researcher in the field of WASH. MUA and SDG (MSS, male, research officer) provided three days of training to the six data collectors (three male and three female WVB-CXB staff who have experience conducting participatory discussions with children) on the background and purpose of the research and the principles of qualitative data collection, including the data collection techniques of KIIs, IDIs, the SB methodology and the sanitation survey, and ethical considerations.

**Table 1. Overview of participant groups by data collection method.**

| Participant Group | Data Collection Method | Site | Number of Activities | Male Participants | Female Participants | Total Participants |
|---|---|---|---|---|---|---|
| Boys aged five to seven years old | Story Book session | 1 | 1 | 6 | 0 | 6 |
| | | 2 | 1 | 6 | 0 | 6 |
| Girls aged five to seven years old | Story Book session | 1 | 1 | 0 | 6 | 6 |
| | | 2 | 1 | 0 | 6 | 6 |
| Boys aged eight to eleven years old | Story Book session | 1 | 1 | 6 | 0 | 6 |
| | | 2 | 1 | 6 | 0 | 6 |
| Girls aged eight to eleven years old | Story Book session | 1 | 1 | 0 | 6 | 6 |
| | | 2 | 1 | 0 | 6 | 6 |
| Caregivers | IDI | 1 | 12 | 0 | 12 | 12 |
| | | 2 | 12 | 0 | 12 | 12 |
| Teachers | KII | 2 | 2 | 0 | 2 | 2 |
| | | 1 | 1 | 0 | 1 | 1 |
| Community Health Workers (CHW) | KII | 1 | 2 | 2 | 0 | 2 |
| Camp Leaders (Majhi) | KII | 1 | 1 | 1 | 0 | 1 |
| | | 2 | 2 | 2 | 0 | 2 |
| Religious Leaders | KII | 2 | 1 | 1 | 0 | 1 |
| Traditional Leaders | KII | 1 | 1 | 1 | 0 | 1 |
| | | 2 | 1 | 1 | 0 | 1 |
| Camp Child Protection Officials (CPO) | KII | 2 | 2 | 1 | 1 | 2 |
| WASH Specialists | KII | 1 | 1 | 1 | 0 | 1 |
| | | 2 | 1 | 1 | 0 | 1 |

## Data collection

SDG and the WVB-CXB staff first piloted the SB methodology with community members, to ensure it was appropriate for use with children. As it was deemed acceptable and provided valuable insights, SDG and the WVB-CXB staff collected data in the local language/s from October 10 to October 25, 2021. The SB sessions were facilitated by a data collector of the same gender as the children and took place in school classrooms; caregivers were sometimes visibly present but not within earshot of the children's discussions. The SB sessions took between 50 and 155 minutes. Caregivers were interviewed in their households, as per their preference. The KIIs took place at the individuals' work/living place, including formal workplaces, households, schools, and mosques, based on their preference. The IDIs and KIIs were between 30 and 60 minutes long. All SB sessions, IDIs and KIIs were audio recorded, and photographs of children's drawings taken during SB sessions. We also surveyed the sanitation facilities available to the children of the caregivers we interviewed. These children and their family members used these sanitation facilities on a daily basis. We visited the available communal sanitation facilities and assessed the status of accessibility issues, toilet walls, toilet roof, toilet door, door handle, door lock, handwashing water container, soap, hygiene promotion and available facilities within the toilets, using an observational checklist [30] (available from www.doi.org/10.17605/OSF.IO/E3KHV).

## Ethical considerations

Self-wetting is a very sensitive topic, and from the outset we had to consider whether it was ethical to conduct such research at a10ll in an emergency setting, particularly the SB sessions which involved children. For an in-depth discussion on our considerations and eventual

decision to conduct the work see S1 Text. Approval to conduct the project was granted by the Research Ethics Committee, Faculty of Engineering, University of Leeds, United Kingdom (Reference MEEC 19–020). Approval to conduct the research in Cox's Bazar was granted by the Institutional Review Board of the Institute of Health Economics (University of Dhaka, Bangladesh), with authority to access the refugee camps granted by the Office of the Refugee Relief and Repatriation Commissioner (RRRC). The research team explained to all participants that they wanted to better understand how children aged five to 11 years old experience self-wetting, and whether there are ways to improve these experiences (participant information sheets are available from [30], www.doi.org/10.17605/OSF.IO/E3KHV). We obtained verbal assent from the children for the SB sessions, and written consent from their caregiver. For the KIIs and caregiver IDIs, we obtained written or verbal (where a second data collector acted as a witness and signed the form) informed consent.

## Positionality and reflexivity

Data was collected by individuals who do not reside within the camps, but who were familiar to the participants through their work for WVB-CXB. This was appropriate as participants were likely more open to discussing such a sensitive topic with outsiders in their professional capacity, as there would be less chance of negative repercussions from disclosure, than if data collectors had been recruited from within the camps. Children and their caregivers appeared comfortable with data collectors and spoke openly about the challenges they face regarding self-wetting.

We endeavoured to uncover both emic and etic perspectives through the research methods. An etic approach allowed us to understand how the experiences of children and caregivers in this specific context may be similar to other humanitarian contexts and human experiences of self-wetting generally, and an emic approach allowed us to interrogate how self-wetting is experienced within the local culture and religion of the camps (a crucial consideration given that societal shame is a common aspect of self-wetting experiences). Using this mixed approach, we could identify aspects of the self-wetting experience which are likely context specific and those which may be universal, as well as practical measures which may be taken by humanitarian practitioners to improve experiences in this specific context, or which may be applicable more generally.

Data collectors did not take formal fieldnotes, but at the end of each day of data collection, the research team conducted a debrief session to critically reflect on the process and adjust the methods as necessary, for example, potential probes for SB sessions and interviews that could be used to identify and explore emerging themes. The team also discussed any verbal, nonverbal, and environmental slights, snubs, or insults, whether intentional or unintentional, which data collectors may be using, which may be communicating negative messages about those who self-wet to children or their caregivers. The team members then discussed how to avoid those unintentional biases and microaggressions and put this into practice in subsequent days.

## Data analysis

All activities were transcribed and translated verbatim into English; participants were not asked to comment on or correct them. MUA and SDG undertook initial data analysis during the evenings of the data collection stage.

After the completion of data collection, following a deductive approach, MUA, SDG, DJB and CRS initially developed a coding framework based on the research objectives. Inductive codes were developed using constant comparative analysis as the work progressed. SDG, AHR, RN, and MAR coded all transcripts using NVivo 12 (QSR International), dividing them

among themselves equally, and coding of all transcripts was then checked by DMS and MUA. Data were triangulated between the data collection methods. Although it was originally planned that initial findings would be discussed with participants to incorporate their feedback into a final round of analysis, restrictions related to COVID-19 and the funding period meant this was not possible.

## Results

The SB sessions, IDIs, KIIs and sanitation survey provided insight into three major facets of self-wetting by children in this humanitarian context: the perceived causes of self-wetting, experiences of children and caregivers when managing the condition, and suggestions for reducing the incidence of self-wetting (See S1 Table for the full codebook).

### Perceived causes of self-wetting in children

The perceived causes of self-wetting in children discussed by participants are summarised in Table 2.

**Self-wetting as a 'disease' or a normal part of life.** Participants indicated that the Rohingya community usually refer to self-wetting as "*Korai*", and it is common among children aged five to 11 years. Several participants consider self-wetting a 'disease' because they believe it needs treatment to be cured. Those caregivers who consider it a disease do so because children urinate in their sleep on a daily basis and continue to do so even after consulting with doctors or *Hakeem* (a herbal medicine practitioner, specially of Unani medicine). A CHW stated that "We call it [self-wetting] a disease because [we can see] when people or children sleep during the daytime and dream, they sometimes defecate/urinate in their sleep. Who [children] cannot control their urge to defecate, whether the toilet is far away or near, they urinate in bed or on clothes when they cannot control it. Others lose control on the way [to the toilets] because the bathroom is far away, unable to hold it all the way." (CHW 2) and one caregiver "If the child is sick, feels troubled, and urinates in bed, won't the mother feel troubled? He [the boy child] is urinating in bed because he has a disease." (Caregiver of Child 3 from SB Session 1).

**Table 2. Perceived causes of self-wetting by children suggested by participants.**

| Overarching theme | Key finding | Respondents supporting key finding | | |
|---|---|---|---|---|
| | | Storybook sessions (n = 8, 48 children) | Caregiver interviews (n = 24) | Key informant interviews (n = 18) |
| Self-wetting as a medical condition | Self-wetting is a disease that needs medical intervention | - | 6 | 5 |
| | Self-wetting is normal and will resolve with age | - | 5 | 10 |
| Self-wetting whilst asleep | Children self-wet when asleep as they are dreaming of urinating | - | - | 3 |
| | Children self-wet whilst asleep as they lack control | 4 | - | - |
| | Children self-wet whilst asleep because they drink too much before bed | 8 | - | - |
| Children are afraid to use the toilet at night | Because it is dark | - | 6 | - |
| | Because the toilets are far away | 5 | 12 | - |
| | Because they are dirty | - | 5 | - |
| | Because the roads to the toilet are in a poor condition | - | 10 | - |
| Children do not use the toilets when they reach them | Because the toilets are not child-friendly | - | 13 | - |
| | Because the line is long and they urinate before it is their turn | - | 9 | 4 |

A few caregivers and several KIs consider self-wetting a normal phenomenon which resolves with age. They believe that as children become older, they gain bladder control; children self-wet when they are still too young to sufficiently control their urine, not because they are ill. One CHW and both Child Protection Officials (CPOs) believe children usually urinate in bed while deep asleep because they are dreaming of urinating in toilets. Children suggested that major reasons their heroes wet themselves at night are excessive intake of water before going to sleep and an inability to control the urge to urinate while dreaming. None of the health care service providers, or the CPOs, are informed about the self-wetting issues of children as the issue is not reported to them by the community.

## Available toilets are inappropriate

The most common reason participants mentioned for children self-wetting was that children consider the toilets available to them are inappropriate for their needs. Some of the caregivers mentioned that children are afraid to use the toilet at night because the toilets are dark, far away, dirty, and the roads to the toilets are in a poor condition. A major reason children gave for their hero self-wetting was distance from the toilet. From the sanitation survey, it was evident that the communal toilets available to the children participants are not all appropriate. We only deemed 4 of the 24 toilets 'child-friendly'. Nine of the toilets are missing door handles and 20 do not have locks. Caregivers reported that the toilets are also frequently broken and the doors need to be closed using wire, which discourages children. As a result of the many inappropriate aspects of the sanitation available to them, children often urinate in their bed or on themselves or their mat. The sanitation survey also revealed that the majority of the toilets are far from households, taking five to ten minutes to get to, and require crossing a sloppy, muddy road. There is no signage to ten of the toilets and difficult-to-see signage for the other 14. Most of the paths (23 of 24) to the toilets do not have any lighting.

In all the SB sessions the children indicated that their hero did not feel comfortable and safe using the toilet. One girl aged eight to 11 years explained that their hero wet the bed because "The latrine is far away; it is scary to go there. That is why it [hero] is late to go there." (SB Session 6). Caregivers stated that "Child feels scared to go to the bathroom at night; there are no lights. The road to the toilet is not good either; the road is dark." (Caregiver of Child 4 from SB Session 6) and "The path to the toilet is not friendly. Now where it is, it is pretty low [in position/placement]. Now which [the new toilets] are going to be newly formed, those should be formed in plain land. Then there will be no more difficulties, and the kids will not fall anymore. But the kids are suffering there now". (Caregiver of Child 4 from SB Session 2).

Caregivers observed that even when children can reach the toilet, they often do not use it because they are generally not child-friendly. Children aged five to 11 years cannot sit properly on the adult toilet seat and face problems in using the water for anal cleansing and flushing. One WASH specialist explained that "latrine's size, pan's size, people's structures, 5 feet, 6 feet, like this. But children's body structure is small. If their pan is 34 inches, 8 inches on this side, they cannot sit with two feet on two sides. It becomes difficult. If we make a child sit on an adult's pan, then s/he will not be able to do it properly. If his legs are spread too wide, he won't feel that pressure." (WASH Specialist 1).

Each block in the Rohingya camps has one toilet that is shared by four to five households, resulting in lengthy wait periods that further deter children from using them. Caregivers mentioned that sometimes, children have to wait for their turn to access the toilet while adults are using it, meanwhile urinating as they are unable to hold it. Some of the service providers confirmed that a long waiting time to get to the toilet is one of the significant causes of self-wetting by children. In addition, because of the volume of users, the available toilets are unclean and

**Table 3. (Anticipated) experiences of children and caregivers when children self-wet.**

| Overarching theme | Key finding | Respondents supporting key finding | | |
| --- | --- | --- | --- | --- |
| | | Storybook sessions (n = 8, 48 children) | Caregiver interviews (n = 24) | Key informant interviews (n = 18) |
| Children are likely distressed when they self wet | Children likely feel uncomfortable, angry, scared, tense and embarrassed after self-wetting | 8 | - | - |
| | Children believe they will be scolded by mothers for self-wetting | 7 | - | - |
| | Children believe they may be beaten for self-wetting | 6 | - | - |
| | Children who self-wet at school are ridiculed or punished (sometimes physically) by teachers and classmates | 4 | - | - |
| It is challenging for some caregivers to manage their children's self-wetting | They become anxious | - | 5 | - |
| | They are uncomfortable | - | 4 | - |
| | They are ashamed | - | 2 | - |
| | It is a physical burden to manage self-wetting in children | - | 3 | 4 |
| Caregivers of children who self-wet do/ would seek help | From religious leaders (Moulovi) | - | 19 | - |
| | From doctors | - | 12 | - |
| | Traditional/spiritual healers are prioritised over doctors | - | 9 | - |

children do not want to use them. As two caregivers stated "This latrine becomes very unclean. There are a lot of people here. There are 8–12 people in each family. For this, the latrine becomes very unclean. We have to clean it; it smells bad if we don't clean it. That's why we have to clean it. Children don't want to use it if it's unclean." (Caregiver of Child 4 from SB Session 6) and "I thought it would be nice to have another latrine for the children. We [the adults] use this latrine, [along with] 3–4 families, together. they [children] don't want to go to unclean latrines." (Caregiver of Child 3 from SB Session 5).

## Experiences of self-wetting in children

The experiences, or anticipated experiences, of children self-wetting are summarized in Table 3.

## Children are likely distressed when they self-wet

In all of the SB sessions children indicated that their heroes felt uncomfortable, angry, scared, tense, and embarrassed after self-wetting. It seemed that through their drawings the children were expressing their own feelings about self-wetting (regardless of whether it is something they experience or not). The children often stated that their hero cried and felt distressed after an incident. One of the main reasons their heroes were scared is that they believed their (heroes') mother would scold them because she was upset, furious, sad and/or embarrassed. For example:

> "*Facilitator: How does your heroine feel when her clothes get wet, dear?*
>
> Respondent 1: She feels troubled. And also feels scared that her clothes were wet, her mother would scold her. . .
>
> Respondent 2: She urinates in bed; that's why she feels terrible and ashamed. . ..
>
> Respondent 3: She feels sad."

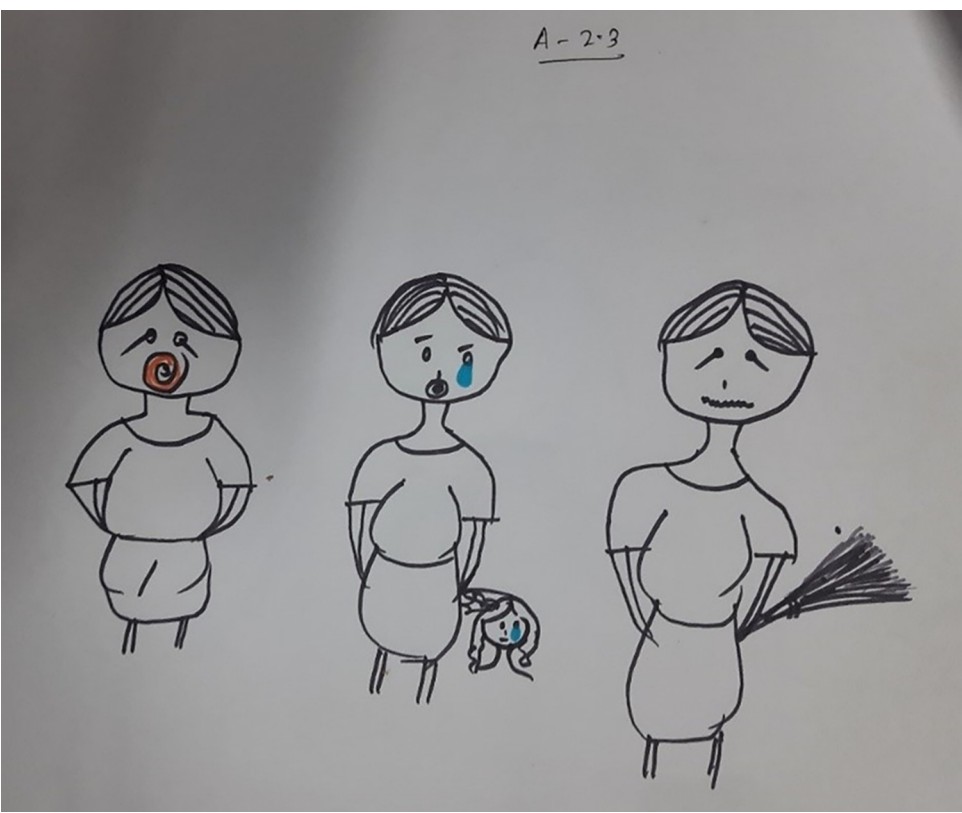

**Fig 1. Drawing of the hero's mother scolding the child, crying and beating the child with a broom, as well as the hero crying.** Story Book Session 6, Girls 8 to 11 Years Old, Site 01 (Photograph by Sudipta Das Gupta).

(SB Session 6)

The children particularly illustrated this when discussing or drawing what would happen when their heroes got out of bed in the morning and had wet themselves. They indicated that caregivers sometimes slap the children or beat them with whatever they can find nearby, such as a broom or stick, or by grabbing their hair (Figs 1 and 2). Some children chose to disclose that this had happened to them.

> "*Facilitator: Suppose one day, the heroine accidentally urinated and wet the bed. How will she feel*?
>
> *Respondent 3: She will be ashamed.*
>
> *Respondent 1: She will be scared.*
>
> *Facilitator: You said that the heroine does not feel good; she cries, she feels scared and sad. Why does she feel like this?*
>
> *Respondent 1: Her mother will beat her."*

(SB Session 3)

In half of the SB sessions children mentioned that teachers and classmates mock and ridicule children who self-wet during school. Children explained that sometimes teachers are

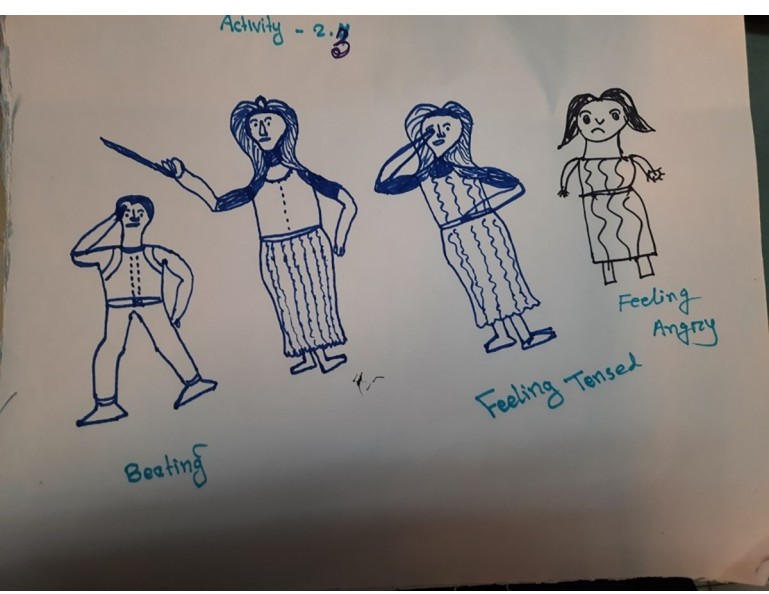

**Fig 2. Drawing of a mother hitting the hero for self-wetting, being tense and being angry.** Story Book Session 1, Boys five to eight years old, Site 2 (English words added by data collector from description provided by illustrator) (Photograph by Sudipta Das Gupta).

compassionate and advise students to go home and change their clothes after urinating in them, but others frequently become angry and beat the students for making their clothes wet at school. They either remove those children from school or call their fathers.

> "*Facilitator: She (heroine) urinated after going to school. What will everyone in the school do? What will the teacher do?*
>
> *Respondent 1: They will make her feel ashamed.*"
>
> *(SB Session 6)*
>
> "*Respondent 1: The teacher beat him (hero).*
>
> *Facilitator: The teacher beat him. Didn't the teacher do anything else?*
>
> *Respondent 3: The teacher drove him away. The teacher kicked him out of school.*"
>
> *(SB Session 7)*

## It is challenging for some caregivers to manage their children's self-wetting

Dealing with their children's self-wetting issues makes some caregivers anxious, uncomfortable, and ashamed. When children urinate in their clothes or bedding, caregivers (particularly mothers) are normally the ones who wash the clothes and linen, which annoys them and sometimes results in punishment of the children; "My son is urinating and defecating; I have trouble washing their clothes. I beat the children and changed their clothes. I feel sad about that." (Caregiver of Child 5 from SB Session 6). In the community, children who experience self-wetting and their parents feel ashamed and face humiliation from the community because of their children's open urination.

Some caregivers mentioned the burden of managing their children's self-wetting. They have to fetch water from a distance to clean the clothes and the child because there is no well

near their homes. Key informants also mentioned that the lack of water availability nearby likely contributed to this stress. Several service providers mentioned that caregivers face challenges in managing children's self-wetting as they have a limited amount of clothes, pillows, and blankets, and if anyone urinates on the beds, they face problems using and cleaning these.

### Caregivers of children who self-wet seek help

To treat self-wetting, caregivers usually do or would seek help, either from religious leaders (Moulovi) or doctors. Many caregivers prioritize traditional/spiritual healers over doctors. The spiritual healers or 'Moulovis' recite the Quran, provide amulets, sacred water, or oil to treat this incontinence issue. As one caregiver explained "Parents have two ideas [when observing incontinence]. I think it would be better to go to the doctor first. [However,] another idea is that [going to the religious leaders]. They believe it happens because of dreams or supernatural jinn ghosts." (Caregiver of Child 3 from SB Session 1).

### Suggestions for reducing self-wetting

Children, caregivers and key informants believe self-wetting can be prevented. Their suggestions for prevention are summarised in Table 4.

### Participants believe that self-wetting can be prevented

A few of the caregivers felt that self-wetting could be prevented if there was better access to doctors' treatments in the camps. The children suggested that to prevent self-wetting their heroes could drink less water and/or urinate before going to sleep at night.

The majority of other recommendations of participants were around improving WASH facilities. Caregivers and key informants suggested that to decrease self-wetting more toilets should be built in general, and these should be well built, nearer to households, have water available and the roads or trails to them improved, with availability of lights at night. A CHW and CPO suggested providing torches for children to use at night when walking to the communal toilets.

Children in half of the SB sessions felt that more toilets needed to be built closer to or in homes. Some caregivers and two Majhi suggested providing a small area at the household for the children to urinate and defecate, "Even if you cannot arrange a bathroom, there should be

**Table 4. Participants suggestions for reducing self-wetting in children.**

| Overarching theme | Key finding | Storybook sessions (n = 8, 48 children) | Caregiver interviews (n = 24) | Key informant interviews (n = 18) |
|---|---|---|---|---|
| Self-wetting can be prevented | If there was better access to doctors | - | 4 | - |
| | If children drank less water | 4 | - | - |
| | If children urinated before going to bed | 4 | - | - |
| | If there were more toilets | - | 4 | 9 |
| | If the toilets were better built | - | 6 | 7 |
| | If the toilets were closer to households | 4 | 6 | 9 |
| | If there was running water available | - | 3 | 8 |
| | If the roads/trails to toilets were improved | - | 3 | 8 |
| | If they were better lit at night | - | 6 | 8 |
| | If children were provided torches | - | - | 2 |
| | If children had an area at the household where they could urinate | - | 4 | 2 |
| | If smaller toilets were provided | - | - | 2 |

a small place for the children (at their house) to urinate and defecate. For example, if the older adults cannot go out, their toilet chairs are arranged." (School Teacher 3). Both WASH specialists suggested that smaller commodes (e.g., smaller pan) should be provided for children so that they can use toilets more easily; "small pans need to be done for children so that s/he can sit comfortably." (Wash Specialist 1)

## Discussion

Our study echoes findings on the impacts of self-wetting documented elsewhere. Self-wetting can lead to negative physical [6], social [2,3] and mental [5,7] health impacts, including that children who self-wet fear, and sometimes experience, verbal and physical abuse from caregivers and teachers [5,8,31]. The stigma associated with self-wetting results in embarrassment and shame, which discourages children from participating in educational and social activities [32,33].

As discussed in the Introduction, there are a range of physiological and psychological causes of self-wetting, and experiences of self-wetting will be influenced by the physical, social and cultural environment a child and their caregiver reside within, as well as individual biological factors. In this small study we asked participants to provide details of their experiences, as well as their insights on the causes of these experiences and their eventual impacts on wellbeing. This likely did not encompass the entire reality of how life within an emergency context affects self-wetting. However, based upon the responses of participants we have developed a theoretical model that indicates some of the factors which contribute to negative experiences and wellbeing impacts (Fig 3). Although WASH and protection practitioners do not have the qualifications to prevent or treat the medical condition of UI, there are three 'pain points' in the model where we believe these practitioners can remove or reduce factors which contribute to poor well-being, thus improving the experiences of children who self-wet and their caregivers. These are improving WASH facilities, providing continence management supplies and

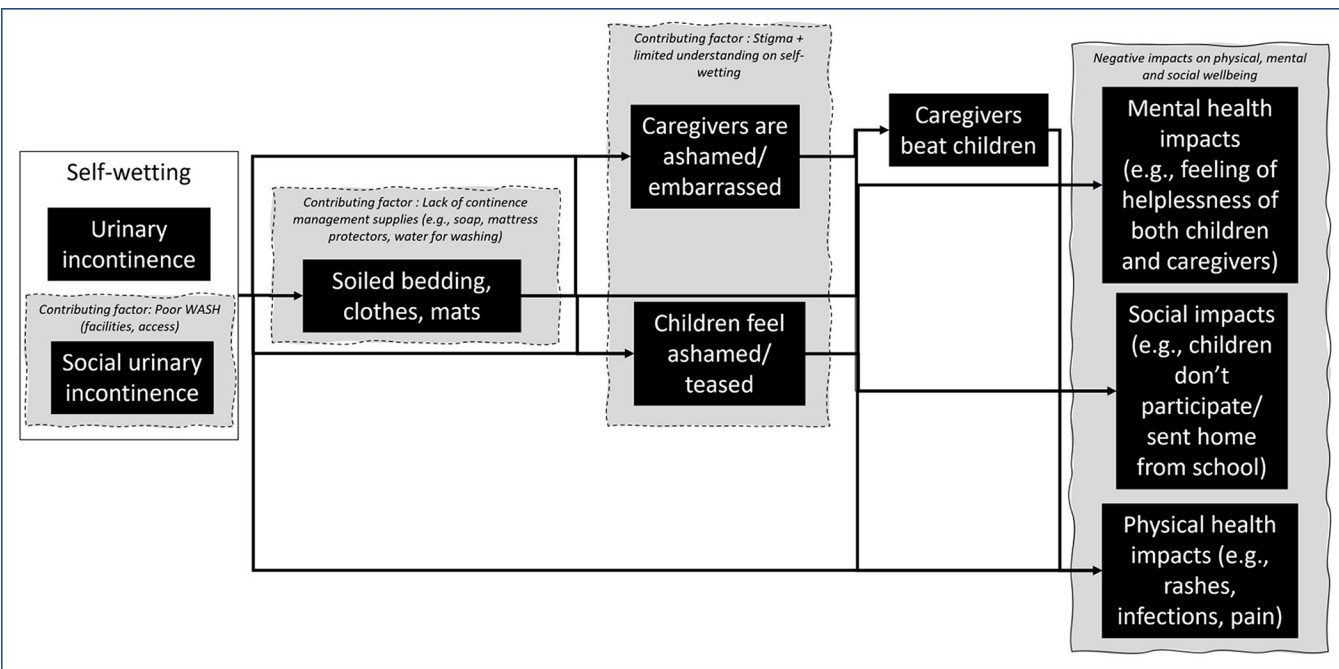

**Fig 3. Theoretical model of self-wetting experiences of children aged 5–11 years in Rohingya refugee camps.**

increasing knowledge of self-wetting whilst reducing stigma. The model could likely be expanded to include more factors which contribute to negative experiences and impacts as further studies are conducted, providing further guidance on how experiences of children and caregivers could be improved.

## Water, sanitation and hygiene facilities

The accessibility issues at and leading to the communal ablution blocks contribute to children experiencing self-wetting, as well as general poor experiences of sanitation even for children who do not self-wet. There are not enough facilities and they have not been designed with children in mind. Similar findings were observed by Ullah [34], who reported this over a decade ago in older sections of the Rohingya refugee camps.

Studies elsewhere have investigated links between children's SUI and toileting infrastructure. These have indicated that increasing the number of bathrooms, establishing child-friendly toilets, ensuring water availability and improving accessibility on the approach to ablution blocks can encourage children to use facilities, thus reducing SUI. For example, a study in the USA found that children who do not have accessible toilets at school are 2.2 times more likely to experience SUI than children to have access to accessible toilets [35]. A study in Kenya showed that improvement to school toilet facilities increased the daytime toilet use of children [36]. Water, sanitation and hygiene practitioners are likely able to improve experiences of all children by installing more WASH facilities that are child-friendly [29].

## Availability of continence management supplies and washing facilities

Caregivers are busy managing children's self-wetting and have to fetch water from a distance to wash clothes and bedding, making them feel depressed, irritated, bothered and restless [17]. In our study, the limited availability of clothes, blankets and diapers make it more challenging for caregivers to manage the leakage issues of children. There is a need for humanitarian actors to more thoughtfully provide non-food items such as mattress protectors, portable toilets for children, and extra soap to the families who have children who self-wet. Médecins Sans Frontières, IRC and IFRC have found the provision of such materials useful to caregivers of children who self-wet in Syria, Iraq, Greece and Honduras [16].

## Knowledge on self-wetting and stigma

The diverse, socio-culturally-determined conceptualization and perception of self-wetting and incontinence impact their management in Rohingya camps. Incontinence is considered a disease among the Rohingya community because the leakage of urine while sleeping is evident even after the consultation with doctors or *Hakeem*. However, prevalence of self-wetting declines with age, with spontaneous cure rates of about 15% yearly between 7 and 12 years and 11% annually between 12 and 17 years [12,37], so medical or *Hakeem* intervention is not always necessary. However, in emergency settings and LMICs, knowledge about incontinence of both caregivers and health workers is still at an introductory level [3,16]. We also found that caregivers often do not report self-wetting issues to health and other support workers due to social stigma [17]. Where they are able to hide self-wetting, children often do not inform their caregivers, as has been observed elsewhere [8].

To reduce stigma, physical and mental abuse towards children, improve the understanding of continence issues, and prioritise useful interventions by humanitarian actors, there is a need to focus on providing more knowledge on self-wetting to caregivers, children, communities and professionals. A study in Brazil found that caregivers with more education are less likely to severely punish children who self-wet [38]. As a result of this research project, we have

developed a contextually appropriate, editable poster that can be displayed in shared public spaces (e.g., child-friendly spaces in the Rohingya camps), explaining that most children will grow out of self-wetting, and should be supported in the meantime (Fig 4). The poster was designed to represent Rohingya individuals but can be used in other contexts where the depictions of individuals are considered appropriate.

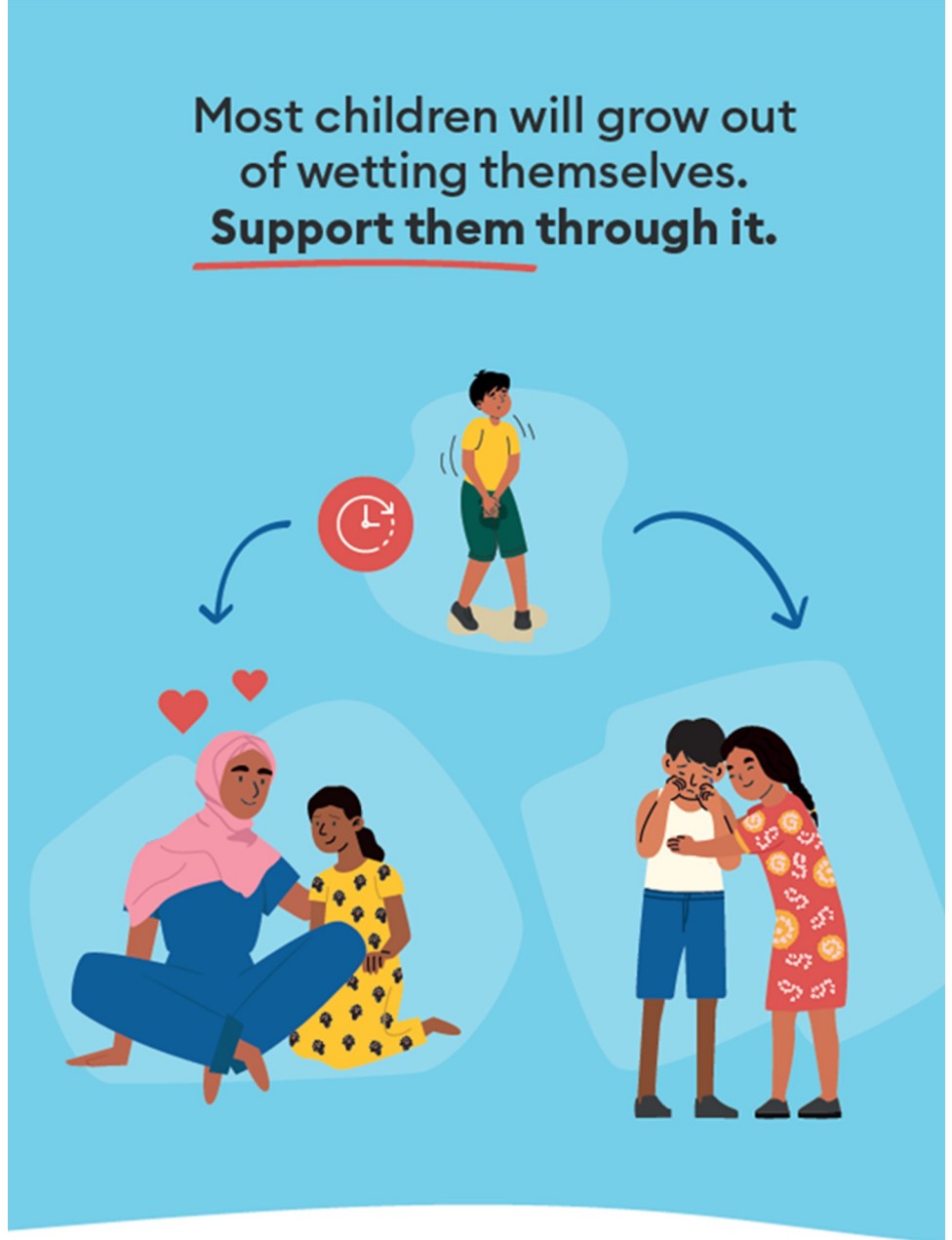

**Fig 4. English version of poster designed to educate the public that most children will grow out of self-wetting and should be supported.** The posters are editable, so the text can be changed to any language, and other text can be added as necessary. The white space at the bottom is intended for local organisations to add their own contact details if viewers would like to know about self-wetting. An editable version of the poster is available from [30] (www.doi.org/10.17605/OSF.IO/E3KHV).

Humanitarian professionals across the globe have called for further knowledge and training on how to address continence issues in emergency contexts [16]. Since the completion of this project, we have worked with Oxfam and the Research Grants Institute of Ghana to develop a training package for humanitarian practitioners based on our combined research on self-wetting in Bangladesh, Ethiopia, Malawi, Uganda and Ghana, as well as existing literature. The training package educates humanitarian practitioners on self-wetting and how simple WASH and protection interventions may assist in improving the experiences of people who self-wet of all ages, and their caregivers [39]. This training package is currently being trialed by Oxfam in 11 different humanitarian situations to understand its applicability in a range of contexts.

## Limitations

Our study only included children and their caregivers in two Rohingya camps. However, the camps we chose broadly represented the living situation of most of the Rohingya community, and we also conducted interviews with diverse stakeholders who have a robust understanding of the situation across the Cox's Bazar camps. Our study did not investigate medical diagnosis or treatment of children with incontinence, and thus did not include health specialists beyond CHWs.

## Conclusion

Children living in the Rohingya refugee camps experience self-wetting. This study is the first of its kind to speak to both children and their caregivers about this issue, eliciting valuable information on their experiences, as well as suggestions for practical changes. Although protection and WASH professionals in emergency settings are not able to prevent or treat the medical condition of urinary incontinence, they can better support children (including those who do not self-wet) and their caregivers through the provision of accessible, close-to-household child-friendly sanitation, providing extra continence management materials such as soap, pads/nappies and mattress protectors, and by both upskilling in their own knowledge around continence issues and communicating this to communities, assisting in better understanding and stigma reduction.

## Supporting information

**S1 Checklist. COREQ (COnsolidated criteria for REporting Qualitative research) checklist.**
(PDF)

**S1 Text. Extracts from the PhD thesis of C. Rosato-Scott detailing development of the Story Book Methodology.**
(DOCX)

**S1 Table. Full codebook.**
(DOCX)

**S1 Questionnaire. PLOS inclusivity in global research questionnaire.**
(DOCX)

## Acknowledgments

The research team acknowledges the many individuals and organisations who contributed to this project, particularly research trainers and data collectors: Hasina Akter, Md. Ibrahim,

Rifat Alam Bristy, Tania Biswash, Mohammad Sawkatur Rahman, Shabnaj Benozir, Rumina Zannat Runa, Hasan Mahmud Rappy, Fahmida Akter. The team also would like to acknowledge the contributions from administrative staff at all organisations, ethical application reviewers and the project's Advisory Board. Most importantly they would like to thank the children who participated in the Story Book sessions and their caregivers.

## Author Contributions

**Conceptualization:** Mahbub-Ul Alam, Claire Rosato-Scott, Dani J. Barrington.

**Data curation:** Mahbub-Ul Alam, Sudipta Das Gupta.

**Formal analysis:** Mahbub-Ul Alam, Sudipta Das Gupta, Asmaul Husna Ritu, Rifat Nowshin, Md Assaduzzaman Rahat.

**Funding acquisition:** Claire Rosato-Scott, Joanne Rose, Barbara E. Evans, Dani J. Barrington.

**Investigation:** Mahbub-Ul Alam, Sudipta Das Gupta, Claire Rosato-Scott, Dani J. Barrington.

**Methodology:** Claire Rosato-Scott, Joanne Rose, Barbara E. Evans, Dani J. Barrington.

**Project administration:** Mahbub-Ul Alam, Nowshad Akram, Joanne Rose, Barbara E. Evans, Dani J. Barrington.

**Supervision:** Mahbub-Ul Alam, Claire Rosato-Scott, Nowshad Akram, Joanne Rose, Barbara E. Evans, Dani J. Barrington.

**Validation:** Mahbub-Ul Alam, Dani J. Barrington.

**Writing – original draft:** Mahbub-Ul Alam, Dewan Muhammad Shoaib, Dani J. Barrington.

**Writing – review & editing:** Mahbub-Ul Alam, Sudipta Das Gupta, Claire Rosato-Scott, Dewan Muhammad Shoaib, Asmaul Husna Ritu, Rifat Nowshin, Md Assaduzzaman Rahat, Dani J. Barrington.

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
