## [Decision Letter · Decision Letter 0]

12 Sep 2023

PGPH-D-23-01619

Experiences of children's self-wetting (including incontinence) in Cox's Bazar's Rohingya refugee camps, Bangladesh

Dear Dr. Barrington,

Thank you for submitting your manuscript to PLOS Global Public Health. After careful consideration, we feel that it has merit but does not fully meet PLOS Global Public Health’s publication criteria as it currently stands. Therefore, we invite you to submit a revised version of the manuscript that addresses the points raised during the review process.

As you will see from the included individual reviews and comments, both reviewers found that there were substantial contributions to knowledge for humanitarian settings. Each reviewer made a different recommendation for publication, and both made recommendations that can seem contradictory. Having reviewed the manuscript and Reviewer 1 and Reviewer 2's feedback, I believe that the manuscript has significant merit and can be made ready for publication with major revisions. In particular, we recommend that the researchers address Reviewer 1's comments, particularly (a) providing additional information about the research team, and (b) the last paragraph, which calls for greater data clarity and a discussion of policy implications. We recommend that you pay careful attention to the totality of Reviewer 2's comments, which have substantive insights for defining the problem, providing context, clarifying data, etc. Please note that you should position your manuscript towards the audience that you feel is most appropriate.

We look forward to receiving your revised manuscript.

Kind regards,

Sharon Alane Abramowitz, Ph.D.

Academic Editor

Journal Requirements:

1. Please provide separate figure files in .tif or .eps format only and remove any figures embedded in your manuscript file. Please also ensure all files are under our size limit of 10MB.

Additional Editor Comments (if provided):

Reviewers' comments:

Reviewer's Responses to Questions

**Comments to the Author**

1. Does this manuscript meet PLOS Global Public Health’s publication criteria? Is the manuscript technically sound, and do the data support the conclusions? The manuscript must describe methodologically and ethically rigorous research with conclusions that are appropriately drawn based on the data presented.

Reviewer #1: Partly

Reviewer #2: Partly

2. Has the statistical analysis been performed appropriately and rigorously?

Reviewer #1: N/A

Reviewer #2: N/A

3. Have the authors made all data underlying the findings in their manuscript fully available (please refer to the Data Availability Statement at the start of the manuscript PDF file)?

Reviewer #1: Yes

Reviewer #2: Yes

4. Is the manuscript presented in an intelligible fashion and written in standard English?

Reviewer #1: Yes

Reviewer #2: Yes

5. Review Comments to the Author

Reviewer #1: The manuscript appears to be technically sound, and the data collection methods used are well-described. The authors conducted interviews and Story Book sessions, which allowed them to gather a range of perspectives from children, caregivers, and other community members in Rohingya refugee camps. Ethical considerations, have been appropriately discussed, indicating a rigorous approach to research ethics. However, I would like to see a small section on the epistemology of the research and background discussions as to why this perspective was necessary in conducting this field work. Also, whether the fidelity to original epistemology was maintained throughout or was subjected to change due to some unanticipated stimuli. Please explain.

Please also add a positionality statement that discusses how the researchers mitigated unconscious bias and microaggression with sub-sections on reflexivity. Additionally, the paper should also present a debate on Etic Vs. Emic perspectives, given that all the researchers in the camps were outsiders. How did camp residents view the researchers? What challenges did this pose? How were these challenges mitigated? The study can also benefit from usage of a theoretical framework. Please revisit your results and see if the ese findings can be linked to potential theories (embodiment/minority stress/rational choice theory etc.).

The data presented in the manuscript appear to support the conclusions drawn by the authors. The findings, including the negative physical, social, and mental health impacts of self-wetting on children and caregivers, are well-supported by the qualitative data collected during the study. Participants provided insights into the causes and consequences of self-wetting, as well as suggestions for potential solutions. If this study has made use of data enumerators from the local region (who are not authors), please acknowledge them one-by-one.

However, the article could benefit from a more structured presentation of the data, potentially using thematic tables or figures to enhance clarity and facilitate a more straightforward understanding of the key findings. Please make use of visuals. Additionally, while the data support the conclusions, the discussion section could further explore the implications of the findings for policy and practice in refugee settings, providing a stronger link between the research and potential interventions.

Reviewer #2: Thank you for the opportunity to review this paper. I was really pleased to see this issue explored in a study – especially as it is so underrecognized by humanitarian actors and there is a lack of evidence on the drivers or solutions in humanitarian settings.

That being said, this paper needs a significant amount of work to address one or some of these gaps effectively. It may be more effective to rework the paper and direct it very clearly and specifically at a WASH audience. The current draft is heavily focused on WASH aspects of incontinence, so reworking it in this way would require a less extensive revision.

I was surprised to read that the introduction doesn’t even mention the role of adversity – especially in humanitarian settings – and hardly mentions the psychological drivers of self-wetting. There are two words in a sentence on page 3 line 74 (“situation-induced trauma, anxiety…” that is in specific reference to humanitarian settings. Otherwise, the paper only refers to physiological and social causes. Incontinence is not a binary issue but is very complex and frequently has psychological drivers and nearly always has psychological consequences and exacerbating factors. This is true for settings experiencing crisis as well as all other settings.

The general living conditions for everyone in the camp – crowding, very basic shelters, fire risk, flooding risk, prevalent interpersonal violence, experiences of caregivers and often of children to torture (witnessing it, surviving it) and extreme violence and subsequent psychological distress or morbidity – these are exceptionally severe conditions. The latrine situation compounds this and has a huge impact on quality of life as well as safety – but the authors focus almost exclusively on the WASH situation rather than the life situation, and they don’t consider how adversity and the responses of different people (parents and teachers beating or shaming children, for example) influence the prevalence of self-wetting, type of self-wetting, experience of it, and children’s as well as adults’ responses to it. This is a major weakness in this paper. The methods are insufficiently described to determine whether or not the study can capture this. However, the authors could triangulate what is known about the bio-psycho-social aetiology of enuresis and encopresis.

Self-wetting seems to be the focus of this paper yet the authors provide evidence on faecal incontinence – which also has huge psychological and social drivers and also different solutions. The term used, “self-wetting” appears to refer to both urinary and faecal incontinence, but this is never made clear. The paper would be improved if the authors explored the findings (if available) for nocturnal enuresis, diurnal enuresis, encopresis (faecal incontinence) and a combination of enuresis and encopresis. Right now it isn’t clear when the authors refer to self-wetting what they refer to. If the study methods don’t allow for the above suggestion, this should be clarified in the text.

The study also found some instances where parents report beating their children and then feeling sad – it is not what makes the parent sad, however the authors appear to attribute the sadness to the child’s incontinence. Another interpretation of the exact quote lines 344-346 is that the mother feels sad about beating the child and about the child’s incontinence. This quote is a very powerful one, and it captures very well the complexity of the issue – children not able to control their bladder or bowels, a parent’s response, a parent’s emotions in the moment and afterwards on reflection. The current analysis doesn’t unpack this, and the result is a valid but oversimplified recommendation for a solution.

The discussion mentions some important consequences of the self-wetting – most notably experiencing multiple forms of violence, withdrawal or absence from school and social activities – but they stop short of describing what this can mean for children’s life trajectories. This is a critical weakness of the paper – exploring this could provide a far stronger argument for improving the WASH conditions for children in the camps.

The authors have developed a causal model for their findings but the methods cannot support such a model. This research can only inform a theoretical model which requires testing.

The focus on self-wetting and solutions for self-wetting in the conclusions in both the text and in the abstract imply that self-wetting is an isolated problem for children. The self-wetting is clearly related to lack of adequate WASH facilities but it is very likely to also be related to experiences of adversity, some of which the authors have described, as well as additional types of severe adversity. The references to encopresis lends further strength to the role of adversity in children’s incontinence in the study sites. Improving latrines would make a huge difference to these children, their families and their communities – but it is unlikely to solve the problem for most children.

Last but not least, the issues presented here – lack of access to adequate sanitation facilities for children, limited access to water, physical abuse, psychological abuse, exclusion from school – these are child rights violations. The way self-wetting is described here suggests that it is very widespread in the study sites. If so, this should be clarified (what is the prevalence? If not known, did the community think self-wetting is common?). The self-wetting is an example of a real, profound impact of these child rights violations not only on the children but on the caregivers and the community. This study lends valuable evidence to support why we should, and must, enable the conditions for children to realise their basic rights.

This paper would be more effective if it were specifically limited to a WASH sector audience – and this should be clearly stated at the beginning and in the abstract. The discussion section on medical interventions should be deleted – it lacks nuance and it is not the focus of the paper or the study, based on what is presented.

Other comments:

What is the estimated prevalence of self-wetting in the camp? Is this known?

Page 3 lines 50-51 – this is a sweeping generalization backed by an outdated reference that is focused on adults including people with physiological incontinence and the elderly. This reference is not relevant for children, especially not for children in humanitarian settings. Please provide sound evidence to back this claim.

Page 3 Line 53 – this is incorrect. Enuresis refers to self-wetting. Wetting during sleep is called nocturnal enuresis. Daytime wetting is called diurnal enuresis, as it usually also occurs during sleep.

It’s great that the authors considered feeding back to the communities, and it is unfortunate that COVID-19 disrupted the plans for this. I am well aware that the restrictions in access to the CXB camps have been particularly challenging, however access has markedly improved during the past couple of years and it is still worth going back to the communities and at the very least sharing the findings, if not internally validating them. This is particularly important for research on sensitive topics with particularly marginalized populations, both for accountability to the population and also to foster trust in the genuine goodwill of researchers. World Vision still has a presence in at least some of the camps. Why were the feedback sessions not adapted to overcome COVID-19 access limitations?

The method of data collection is not even briefly described and the reader is simply given a reference so they can go look it up if they’re interested. Later in the results, children talk about heroes wetting themselves – which will sound very strange to someone not aware of the storybook method. It is not acceptable to expect the reader to do additional reading to be able to understand the study and this paper. A brief description of the method of data collection with children would fix this, in addition the reference for additional reading.

Did the data collection methods have any protocol for what to do if a child disclosed abuse or exhibited psychological distress? Was there any protocol to refer to child protection services? Did sessions continue when children exhibited distress? Was a psychologist available?

The first statement of the results is a sweeping generalization which is not supported by the study design. The authors attribute the findings of a single study in Cox’s Bazar to all humanitarian settings.

What is a Hakeem? This is referred to in a couple of places but never explained.

Page 12 line 279- typos: the family size is presented as a fraction rather than a range.

Page 13 lines 289-291 – the authors make inferences which do not appear to be based on objective findings from the study (e.g. “seem…”)

There is a lot of repetition of the findings in the discussion section. The discussion should not repeat the results section but should explore the meaning of the findings and how this compares to what else is known on the topic.

The conclusion could be much stronger - could improving the toilets do more than improve the lives of children who self-wet? What about the driving factors – and likely impacts on a much larger population? And how would improving the toilets improve other aspects of the children’s and families’ lives?

6. PLOS authors have the option to publish the peer review history of their article (what does this mean?). If published, this will include your full peer review and any attached files.

**Do you want your identity to be public for this peer review?** For information about this choice, including consent withdrawal, please see our Privacy Policy.

Reviewer #1: **Yes: **Ateeb Ahmad Parray

Reviewer #2: No

---

## [Decision Letter · Decision Letter 1]

15 Feb 2024

Experiences of children's self-wetting (including urinary incontinence) in Cox's Bazar's Rohingya refugee camps, Bangladesh

PGPH-D-23-01619R1

Dear Dr Barrington,

We are pleased to inform you that your manuscript 'Experiences of children's self-wetting (including urinary incontinence) in Cox's Bazar's Rohingya refugee camps, Bangladesh' has been provisionally accepted for publication in PLOS Global Public Health.

Best regards,

Julia Robinson

Executive Editor

Reviewer Comments (if any, and for reference):

Reviewer's Responses to Questions

**Comments to the Author**

1. If the authors have adequately addressed your comments raised in a previous round of review and you feel that this manuscript is now acceptable for publication, you may indicate that here to bypass the “Comments to the Author” section, enter your conflict of interest statement in the “Confidential to Editor” section, and submit your "Accept" recommendation.

Reviewer #1: All comments have been addressed

2. Does this manuscript meet PLOS Global Public Health’s publication criteria? Is the manuscript technically sound, and do the data support the conclusions? The manuscript must describe methodologically and ethically rigorous research with conclusions that are appropriately drawn based on the data presented.

Reviewer #1: Yes

3. Has the statistical analysis been performed appropriately and rigorously?

Reviewer #1: N/A

4. Have the authors made all data underlying the findings in their manuscript fully available (please refer to the Data Availability Statement at the start of the manuscript PDF file)?

Reviewer #1: Yes

5. Is the manuscript presented in an intelligible fashion and written in standard English?

Reviewer #1: Yes

6. Review Comments to the Author

Reviewer #1: (No Response)

7. PLOS authors have the option to publish the peer review history of their article (what does this mean?). If published, this will include your full peer review and any attached files.

**Do you want your identity to be public for this peer review?** For information about this choice, including consent withdrawal, please see our Privacy Policy.

Reviewer #1: **Yes: **Ateeb Ahmad Parray
